# Evaluation of the Degree of the Value Realization of Ecological Products of the Forest Ecological Bank in Shunchang County

**Ding Xu** [1,†]**, Yajun Wang** [2,†]**, Lianbei Wu** [1] **and Weimin Zhang** [1,*]

1   School of Economics and Management, Beijing Forestry University, Beijing 100083, China;
    xuding@bjfu.edu.cn (D.X.); wulianbei@bjfu.edu.cn (L.W.)
2   School of Landscape Architecture, Beijing Forestry University, Beijing 100083, China; wyj0418@bjfu.edu.cn
*   Correspondence: zwm@bjfu.edu.cn
†   These authors have contributed equally to this work.

**Abstract:** The forest ecological bank (FEB) plays a vital role in the transformation of ecological assets into ecological capital. The purpose of this study is to clarify the role of Shunchang County's FEB in promoting the realization of the value of forest ecological products and the degree of the value realization of FEB ecological products so as to provide data support and policy reference for sustainable forest management and an ecological product value realization model. The ratio of the sum of the material supply value and the forest premium benefit of the forest ecosystem of the state-owned forest land to the total production value of the forest ecosystem of the state-owned forest land in Shunchang County is taken as a quantitative index of the realization degree of the ecological product value of the FEB in Shunchang County. (1) The difference in the production value of the forest ecosystem per unit area between state-owned forest land and non-state-owned forest land is USD 340.17, and the production value of the ecosystem brought about by the scientific cultivation of the FEB has increased by USD 25.92 million. (2) The base price of state-owned forest land in Shunchang County is USD 378.30, the base price of non-state-owned forest land is USD 247.23, and the value-added premium value of forest land is USD 30.19 million. (3) The realization degree of the ecological product value of the FEB in Shunchang County is 85.51%. These results show that the FEB can accelerate the progress of forest ecological products and play an important role in the construction of the ecological civilization proposed by China.

**Keywords:** forest ecological bank; ecological product value; realization degree evaluation

## 1. Introduction

Forest resources are undergoing profound changes. Growing environmental problems have led to the realization that ecological resources are the material basis for the protection of the ecological environment for sustainable development and the maintenance of life. Forests represent the largest ecosystem on land, as well as the terrestrial ecosystem with the most complex structure, the most functions and the most stable processes on Earth, providing a wide variety of ecological products for human production and life as well as economic and social development [1,2]. China has a large land area which includes many ecosystems, and its ecological resources are rich in quantity and variety [3]. China is building on the concept that "Green hills are golden hills." Its essence is to turn ecological resources into assets and capital [4,5]. A sustainable development mechanism that transforms the "green mountains" of natural resources into "golden mountains" of economic development is an important issue in the governance of ecological civilization at present. In the 1970s, the eco-bank proposed by the Federal Republic of Germany was the beginning of exploring the transformation of natural resources into natural assets [6]. Subsequently, the wetland mitigation banks and forest banks proposed by various countries are all complements and expansions of the eco-banking operation model [7]. In 2018, Nanping City, Fujian

Province, China, created China's first forest ecological bank (FEB), relying on its rich forest resources. It was also promoted as a typical case to the whole country. The establishment of an FEB symbolizes the unity of economic, social, and ecological benefits. Thus, it should be proven that the ecological bank can efficiently utilize ecological resources and promote the realization of the value of ecological products.

An FEB is a typical practice for the unified utilization of scattered forest resources. It is an institution that promotes the development of the ecological economy by managing forest ecological resources, improving forest ecosystem services, and realizing the protection and management of the forest ecosystem. An FEB is a platform for forest resource integration, quality enhancement, and operation, with financial attributes such as forestry financing and guaranteed function, which is constructed by drawing on the "decentralized input, centralized output" model of commercial banks and is also a platform for the implementation of entrepreneurial management [7]. It is also an economic entity with entrepreneurial management and market-oriented operation. Relying on the forestry science and technology advantages, seedling advantages, and scale operation advantages of the county's state-owned forest farms, it improves the output of forest land through the implementation of a forest quality precision enhancement project [8]. At the same time, according to the location advantages and forest conditions, respectively, it develops forest recreation, a forest economy and other ecological projects, enriches the tangible and intangible ecological products in the forest, expanding the new forestry industry, protects the ecological safety of forestry under the premise of reducing operating costs and risks, and maximizes comprehensive benefits [9,10]. The function of an "FEB" includes six aspects: integration, restoration, innovation, transaction, financing, and operation. Based on maintaining the value of ecosystems, it builds a natural resource asset operation and management platform, centralizes the storage and remediation of scattered ecological resources into high-quality asset packages, connects to the capital market, and introduces market-oriented capital and professional operators, thus transforming resources into assets and capital and innovating a multi-principal, market-oriented mechanism for realizing the value of ecological products.

For the study of FEBs, most studies focus on how ecological banks transform natural resources into capital for theoretical research [11]. They also expound the mechanism of FEBs and how stakeholders promote the implementation of FEB policy [12,13]. There are few studies on the specific evaluation of FEBs to improve economic and ecological benefits [14,15], and the current research only accounts for the value of ecological products and theoretically analyzes the path of value realization [16]. To supplement this research, this paper constructs an evaluation framework for the evaluation of the value realization degree of ecological products, and evaluates the value realization degree of the ecological products of Shunchang County's FEB.

Shunchang County, as the first area in China to operate forest resources with an FEB, plays a vital role in the sustainable development of China's economy and ecology. At present, China is making the realization of the value of ecological products an important way to promote sustainable development. It has become an urgent task for government departments to seek how to reasonably evaluate the realization of the value of regional ecological products. An FEB is an effective integration of environmental protection and economic development. The FEB has formed a sustainable mechanism and mode for realizing the value of ecological products. Therefore, we express the two research questions of this study as follows: How can an evaluation model for the value realization degree of ecological products of an FEB be constructed? What are the implications we can obtain from the evaluation results? In this process, we combined the relatively mature ecological service function value accounting method and forest land classification and grading method with the forest resource endowment conditions of Shunchang County, carried out a local optimization and improvement, and adjusted the corresponding key parameters. The degree of value realization we assessed can be an indicator of the local sustainable development process and according to its calculation process, we can clarify the supply capacity of ecological products and economic growth. This not only lays a foundation

for the practical work of a local ecological product value realization mechanism but also provides technical support for the degree evaluation of other ecological product value realization pilot areas.

## 2. Materials and Methods

### 2.1. Study Area

Shunchang County is in the northwestern part of Fujian Province, between a longitude of 117°30′~118°14′ East and a latitude of 26°39′~27°12′ North. Shunchang County is subordinate to Nanping City, adjacent to Jianou in the northeast, Nanping in the southeast, Shaxian in the south, Jiangle in the west, Shaowu in the northwest, and Jianyang in the north. The county is 74 km long from east to west and 61 km wide from north to south, with a total area of 1985 square kilometers (Figure 1).

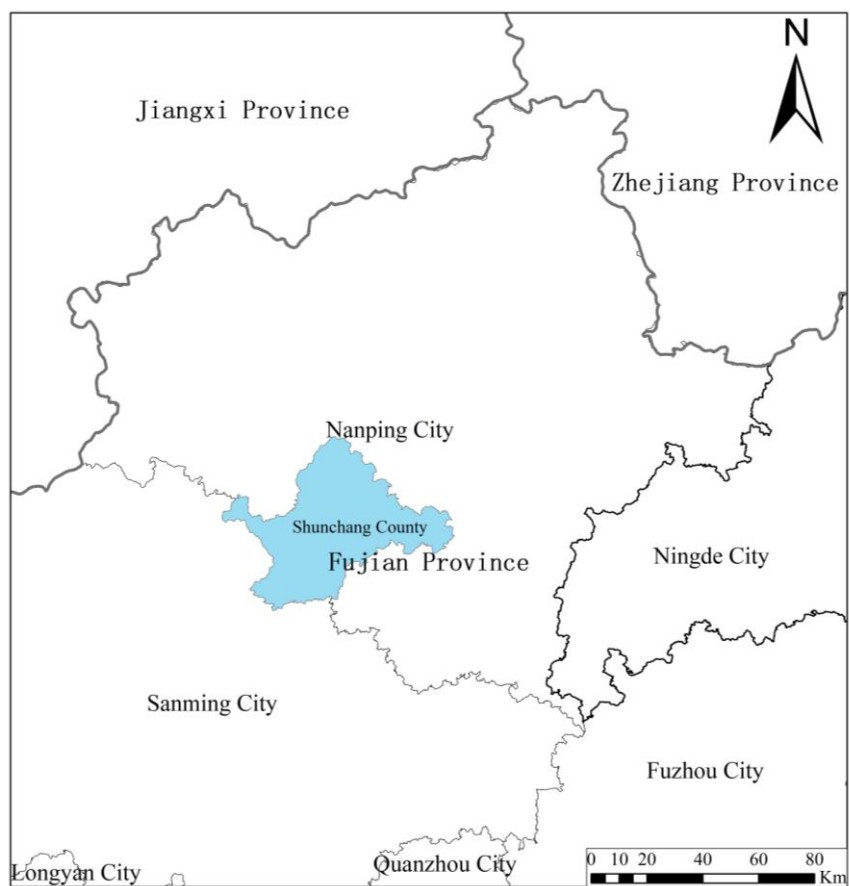

**Figure 1.** Shunchang County administrative district map.

Shunchang is a key forest region in southern China. The county's forest land area is 167,300 hectares, including 70,700 hectares of Chinese fir, 44,000 hectares of bamboo forest, and 29,300 hectares of broad-leaved forest. The forest volume is 15.31 million cubic meters, and the number of bamboo stands is 110 million. It is the only Chinese fir town, the first Chinese bamboo town, and the demonstration county for the national timber strategic reserve base, with a forest coverage rate of 79.8%. There are 187 families, 713 genera and 1399 species of vegetation in Shunchang. Among them, there are 33 families, 58 genera and 106 species of ferns. Regarding Gymnospermae, there are 9 families, 19 genera, and 30 species; there are 145 families, 636 genera, and 1263 species of angiosperms. There are 24 species of nationally protected wild plants and 14 species of provincially protected wild plants. There are 12 general and 78 species of bamboo plants, of which Phyllostachys edulis is the most common. Precious tree species mainly include camphor, nan, sassafras, palm wood, cypress wood, cypress, qinggang oak, cedar, yew, and ginkgo.

*2.2. Framework for Assessing the Degree of the Realization of Ecological Product Values in an FEB*

The realization of the value of ecological products is a hot topic for scholars from all walks of life. However, there are few studies on the degree of the value realization of ecological products, and there is a lack of quantitative analysis of the degree of value realization. For our research subject FEB, the realization of ecological product value in this mode is the process of transforming ecological assets into ecological products and giving full play to their value. By improving the quality of a forest ecosystem, an FEB can effectively increase ecological benefits and form a good ecological environment. The improvement of ecological quality will also promote the development of the local economy and society. Therefore, we will evaluate the degree of realization of the value of ecological products from both ecological and economic perspectives to explore how to influence the regional ecological economy and how to influence the realization of the value of local ecological products after the development of an FEB.

2.2.1. Analysis of Ecological Quality Improvement of the FEB in Shunchang County

The production value accounting of the Shunchang forest ecosystem is based on China's forest ecosystem service function assessment specifications [17,18]. The difference between the production value per unit area of state-owned and non-state-owned forest land, combined with the ecological bank storage value, is the ecological value enhancement impact brought by the operation of Shunchang County's FEB to Shunchang County. The production value of forest ecosystems accounts for the sum of the values of all ecological goods and services provided by ecosystems to human society in each region and is expressed as follows:

$$GEP = V_1 + V_2 + V_3 \tag{1}$$

where $GEP$ is the value of the ecosystem's production; $V_1$ is the value of the ecosystem's provisioning services; $V_2$ is the value of the ecosystem's regulating services; and $V_3$ is the value of the ecosystem's cultural services.

According to whether forest products and services have economic attributes or not, the object of forest product flow accounting is categorized into economic forest products and services and public goods and services.

(1)     Economic forest ecological products:

With reference to the specific items and evaluation indicators for provisioning services and cultural services in the Specification for the Assessment of Forest Ecosystem Service Functions (GBT 38582-2020) [19], it is determined that the accounting indicators for economic forest ecological products and services are timber products and non-timber products.

(2)     Public goods and services of forests:

The public goods and services of forests account mainly for ecological products and services provided by forest resource assets that cannot realize economic benefits through market transactions but can provide various benefits directly to humans, such as most forest regulation services and some forest cultural service products [20].

Accounting indicators for public welfare forest products and services should be final products, and intermediate products should not be included as accounting indicators. In accordance with the principle of human benefits and final products, and with reference to the Specification for the Assessment of Forest Ecosystem Service Functions (GBT 38582-2020) and the specific items and evaluation indexes for regulating services and cultural services in the Technical Specification for the Assessment of Forest Resource Values (LYT1721-2008) [21], it is determined that the accounting indexes of public welfare forest products and services shall be water conservation, air purification, soil retention, carbon sequestration, negative oxygen ion release, and cooling regulation.

2.2.2. Analysis of the Forest Land Premium Effect of the FEB in Shunchang County

Through the study of forest land grading, state-owned and non-state-owned forest land in Shunchang County's forest ecosystem was graded [22,23]. Based on determining the baseline price of forest land separately, the difference between the land prices of the two types of forest land was calculated as the economic impact of the FEB on Shunchang County [24].

Shunchang County's FEB, based on the wishes of forest households, innovatively introduced redemption, share cooperation, leasing, trusteeship and other single or combined ways to centralize the transfer of forestry resources, increasing the options for forest households to transfer forest rights and solving the problems of inefficient development and sloppy operation brought about by the fragmentation of forest rights [25]. And with the help of the state-owned forest farm escrow for forest quality precision improvement, its forest quality, ecological benefits, timber output, and other situations are much higher than the non-state-owned forest land foresters' independent management. In this appraisal, the difference in land value between state-owned forest land and non-state-owned forest land in Shunchang County is taken as the premium to the local forest land value in Shunchang County after the forest eco-banking work was carried out in Shunchang County [26].

The forest land classification methodology is calculated as follows:

(1)     Determination of grading units:

The forest land grading unit is the basis for the calculation of the scores of each grading factor and the basic spatial unit for assessing the grade of forest land, and its internal land characteristics and location conditions are relatively consistent. Small groups in forest resource planning and investigation generally have the same management objectives and are the basic units of forest investigation and management, and their internal structural characteristics are largely the same, and the management measures taken are basically similar. Therefore, small groups are selected as forest land classification units [27].

(i)     Selection of grading factors:

Forest land grading factors should be selected to reflect the value characteristics of forest land, including natural factors, socio-economic factors, and location factors. Natural factors reflect the quality of forest land and are influenced by climate conditions, topography, hydrology, and soil. Since the climatic conditions of the same county (city) are comparable, hydrological conditions may be related to the quality of forest land, but it is difficult to obtain the data accurately. Therefore, based on the principle of significant difference and the easy accessibility of data, and in combination with the results of the existing research, the natural factors determine the indicators of topography and soils, including soil type, soil thickness, humus thickness, slope, slope position, slope direction and elevation; the social factors should include socio-economic conditions, but because of the same county (city), the social factors should be included in the rating. Social factors should include socio-economic conditions, etc. However, since there is not much difference in socio-economic conditions within the same county (city), socio-economic factors are not considered in the classification of forest land within the same county (city); location factors are closely related to forestry production, mainly including the distance of timber transportation and the distance of timber collection [28].

(ii)     Quantification and assignment of indicators:

The entropy value method requires all indicators to have specific values for the calculation of weights; quantitative indicators are directly weighted by their values, while some indicators are qualitative indicators without specific values which must be indirectly quantized. The indicators are classified into different grades, and then the graded values are used for the calculation of weights. The simple weighted sum model is used to calculate the grading index first according to the level of expert scoring to assign different roles to values, and then the weight value and the role of the assigned value are used for the

calculation of the grading index [29]. The results of the quantification and assignment of indicators are shown in Table 1.

**Table 1.** Assignment of roles at the indicator level.

| First-Level Indicators | Secondary Indicators | Index Characteristics | Index Level | Role Assignment |
|---|---|---|---|---|
| Natural factors | Soil texture | Loam | 1 | 8.75 |
| | | Clay | 2 | 5.5 |
| | | Sand | 3 | 2.75 |
| | Soil thickness/cm | >100 | 1 | 9 |
| | | 51~100 | 2 | 7 |
| | | 31~50 | 3 | 5 |
| | | 16~30 | 4 | 3 |
| | | ≤15 | 5 | 1 |
| | Humus thickness/cm | >20 | 1 | 9 |
| | | 15~20 | 2 | 7 |
| | | 10~15 | 3 | 5 |
| | | 5~10 | 4 | 3 |
| | | ≤5 | 5 | 1 |
| | Altitude/m | $H \leq 500$ | 1 | 9 |
| | | $500 < H \leq 800$ | 2 | 7 |
| | | $800 < H \leq 1000$ | 3 | 5 |
| | | $1000 < H \leq 1200$ | 4 | 3 |
| | | $H > 1200$ | 5 | 1 |
| | Slope/(°) | Flat slope, gentle slope (≤15°) | 1 | 9 |
| | | Slope (15°~24°) | 2 | 7 |
| | | Steep slope (25°~34°) | 3 | 5 |
| | | Steep slope (35°~44°) | 4 | 3 |
| | | Dangerous slope (≥45°) | 5 | 1 |
| | Aspect | No slope, north slope | 1 | 9 |
| | | East slope, northeast slope | 2 | 7 |
| | | Northwest slope, southeast slope | 3 | 5 |
| | | West slope | 4 | 3 |
| | | Southwest slope, south slope | 5 | 1 |
| | Slope position | Flat, all slope | 1 | 9 |
| | | Valley, downhill | 2 | 7 |
| | | Middle slope | 3 | 5 |
| | | Uphill | 4 | 3 |
| | | Spine | 5 | 1 |

| First-Level Indicators | Secondary Indicators | Index Characteristics | Index Level | Role Assignment |
|---|---|---|---|---|
| Location factor | Skidding distance/km | Grade I ($\leq$2) | 1 | 8.75 |
| | | Grade II ($2 < S \leq 4$) | 2 | 5.5 |
| | | Grade III ($4 < S \leq 6$) | 3 | 2.75 |
| | Transportation distance/m | $\leq$20 | 1 | 9 |
| | | 21~25 | 2 | 7 |
| | | 26~30 | 3 | 5 |
| | | 31~35 | 4 | 3 |
| | | >35 | 5 | 1 |

(iii)   Methodology for calculating indicator weights:

The entropy value method is used to determine the weights of each indicator, taking into account the existence of positive and negative correlation differences between different indicators, and it has a different outline and unit of measurement, so the indicators are standardized in two ways, and $Y_{ij}$ (i = 1, 2..., m; j = 1, 2..., n) is set as the value of the jth indicator of the ith unit [30]. The normalization of positive indicators is handled as follows:

$$Y_{ij} = \frac{Y_{ij} - min(Y_{ij})}{max(Y_j) - min(Y_j)} \tag{2}$$

Negative indicators normalized are treated as follows:

$$Y_{ij} = \frac{max(Y_j) - Y_{ij}}{max(Y_j) - min(Y_j)} \tag{3}$$

The share of the value of the indicator in the ith cell under the jth indicator is:

$$r_{ij} = \frac{Y_{ij}}{\sum_{i=1}^{m} Y_{ij}} \tag{4}$$

The determination of entropy values for each indicator:

$$P_j = -k \sum_{i=1}^{m} r_{ij} ln(r_{ij}) \tag{5}$$

where $P_j$ is the indicator entropy value; $k = \frac{1}{ln(m)}$, and j = 1, 2..., n. When $Y_{ij} = 0$, $r_{ij} ln(r_{ij}) = 0$. The determination of the coefficient of variation for each indicator:

$$g_j = 1 - P_j$$

where $g_j$ is the indicator's coefficient of variation.
The determination of the weights of the indicators:

$$W_j = \frac{g_j}{\sum g_j} \tag{6}$$

where $W_j$ is the weight value.

(iv)   Calculation of the classification index:

A simple weighted summation model was used to carry out the calculation of the forest land grading index and to develop a grading criterion based on the calculated results. The simple weighted summation model is as follows:

$$P = \sum_{i=1}^{n} W_j * F_j \tag{7}$$

where $P$ is the grading index, $W_j$ is the indicator weight value, and $F_j$ is the indicator role assignment.

(v) Determination of forest land classes:

The forest land grade is divided according to the distribution of the grading index, and different forest land grades are divided according to different score intervals. The grading index obtained via any evaluation unit can only correspond to one forest land grade, and the level of the forest land grade reflects the degree of quality of the forest land (including natural and socio-economic conditions); there should be a gradual transition between grades, and the number of forest land grades is determined according to the different forest lands.

(2) Benchmark valuation of forest land:

The formula is

$$B_u = \frac{A_u + \sum_{j=1}^{n} D_j \times (1+i)^{u-j} - \sum_{j=1}^{n} C_j \times (1+i)^{u-j+1}}{(1+i)^u - 1} - \frac{V}{i} \tag{8}$$

where $B_u$ is the value of forest land, $A_u$ is the net income from the main harvest in the year u of a realistic stand, $D_j$ is the net income from annual harvests in the years j, $C_j$ is the direct investment in forestry in each year, V is the average indirect cost of forest production, $i$ is the interest rate (the interest rate without inflation), and $n$ is the number of years in the rotation period [31].

The specific data sources are a net income of 149.85 USD/year from the main harvest of a realistic stand on state forest land; an income from intercutting of USD 41.36, a direct investment of USD 17.92, and forestry costs of USD 10.34.

2.2.3. Assessment of the Degree of Realization of the Ecological Product Values of the FEB in Shunchang County

We take the ratio of the sum of the material supply value of forest ecosystems on state-owned forest land and the premium benefits of forest land to the total production value of forest ecosystems on state-owned forest land in Shunchang County as a quantitative indicator of the degree of the realization of the value of ecological products of the FEB in Shunchang County. The degree of realizing the ecological product value of Shunchang County's FEB is the degree of promoting the comprehensive development of local economy and ecology in Shunchang County through the implementation of the FEB. Therefore, we believe that the formula means the extent to which the economic benefits of the Shunchang FEB, through the management of state-owned forest resources, are converted into the value of ecological products, and the formula is as follows:

$$De\_RE = (PREM + MP)/GEP\_Value \tag{9}$$

where $De\_RE$ is the degree of the realization of the ecological product value of the FEB in Shunchang County; *PREM* is the value-added premium from the difference in land value between state-owned and non-state-owned forest land; *MP* is the value of material supply of the forest ecosystem of state-owned forest land in Shunchang County; and *GEP_Value* is the production value of forest ecosystems on state-owned forest land in Shunchang County. The present value of the premium is calculated by taking the 25-year logging rotation

period as the period and 5% as the discount rate and calculating the premium in the current year.

## 3. Results

### *3.1. Results of Accounting for the Production Value of Forest Ecosystems in Shunchang County*

After accounting, the production value of forest ecosystems in Shunchang County in 2021 totaled USD 187,846,781.47. Among them, from the perspective of accounting regional distribution, the state-owned value of USD 120,496,243.49 and the non-state-owned value of USD 67,350,537.98, respectively, accounted for 64.15% and 35.85% of the total value of the accounting; from the perspective of accounting subjects, the highest value for the subject of water containment accounted for 80.04%, followed by soil retention, and anion release was the least, representing less than 1% of the total value of the accounting.

The forest ecosystem area of Shunchang County's FEB is 0.51 million hectares, accounting for 21.34% of the state-owned forest land. According to the calculation, the production value of the forest ecosystem in Shunchang County's FEB is USD 37.68 million; of this amount, the value of public welfare products and services is USD 31.17 million, and the value of economic products and services is USD 6.5079 million. The physical quantity and value corresponding to each of the specific accounting subjects are shown in Tables 2 and 3.

**Table 2.** Value and physical quantities of forest public goods and services in Shunchang County.

| Public Goods and Services | Value Quantities (USD) | Physical Quantities | Physical Quantity Unit |
| --- | --- | --- | --- |
| water conservation | 12,563.49 | 1,046,985,084 | $m^3/a$ |
| soil conservation | 1721.78 | 4482.40 | million tons/a |
| air purification | 72.52 | 526,062.55 | kg/a |
| carbon sequestration | 37.85 | 714,501.32 | $t·CO_2/a$ |
| negative oxygen ion release | 5.96 | $1.31 \times 10^{25}$ | pcs/a |
| cooling regulation | 1138.61 | 5,891,172,027 | kW·h/a |

**Table 3.** The physical quantities of forest economic products in Shunchang County.

| Oleaginous Seeds (Tons) | Celandine Seeds (Tons) | Oil Tea Seeds (Tons) | Brown Flakes (Tons) | Turpentine (Tons) | Dried Bamboo Shoots (Tons) | Thatch (Tons) | Tree Economic Forest (Tons) |
| --- | --- | --- | --- | --- | --- | --- | --- |
| 403 | 118 | 1336 | 12 | 47 | 10,365 | 6617 | 6543.39 |

According to the survey data from the third national land survey and the second category of forest resources in Shunchang County, the number of sub-compartments of state-owned forest land in Shunchang County is 4468, the area of state-owned forest land is 16,243.67 hectares, and the production value of forest ecosystem per unit area of state-owned forest land is USD 494.64. The number of sub-compartments of non-state-owned forest land is 8004, the area of non-state-owned forest land is 29,061.4 hectares, and the production value of the forest ecosystem per unit area of non-state-owned forest land is USD 154.47. The difference in the forest ecosystem production value per unit area between state-owned forest land and non-state-owned forest land is USD 340.17. Combined with the collection and storage area of the FEB, it can be concluded that the production value of the ecosystem brought by the scientific cultivation of the FEB has increased by USD 25.92 million.

### *3.2. Analysis of the Forest Land Premium Effect of Shunchang's FEB*

3.2.1. Results of Forest Land Classification

After utilizing the entropy value method to carry out the weight calculation of each index, assigning each index to the forest land for the grading index calculation, and

comparing various forest land grade division methods, the results are suitable for the use of the total score axis to determine the method of obtaining the results of different property rights based on the nature of the forest land grade in Tables 4 and 5.

**Table 4.** Shunchang County state forest land grading results.

| Forest Land Class | Number of Classes | Percent | Area (Hectares) | Percent |
|---|---|---|---|---|
| 1 | 1565 | 37.40% | 55,049.33 | 33.89% |
| 2 | 1066 | 25.30% | 34,393.33 | 21.17% |
| 3 | 1677 | 32.80% | 62,905.33 | 38.73% |
| 4 | 160 | 4.50% | 10,088.67 | 6.21% |

**Table 5.** Shunchang County non-state forest land grading results.

| Forest Land Class | Number of Classes | Percent | Area (Hectares) | Percent |
|---|---|---|---|---|
| 1 | 2564 | 32.02% | 88,718 | 30.53% |
| 2 | 2386 | 29.80% | 70,671.33 | 24.32% |
| 3 | 2847 | 35.56% | 120,162.67 | 41.35% |
| 4 | 210 | 2.62% | 11,062 | 3.81% |

The evaluated forest land was divided into four forest land classes, and the state-owned and individually operated forest land resources were mainly concentrated in classes 1, 2, and 3, accounting for 95.5% and 97.3% of the total number of total forest land classes and 93.7% and 96.1% of the total area of forest land, respectively (Figure 2).

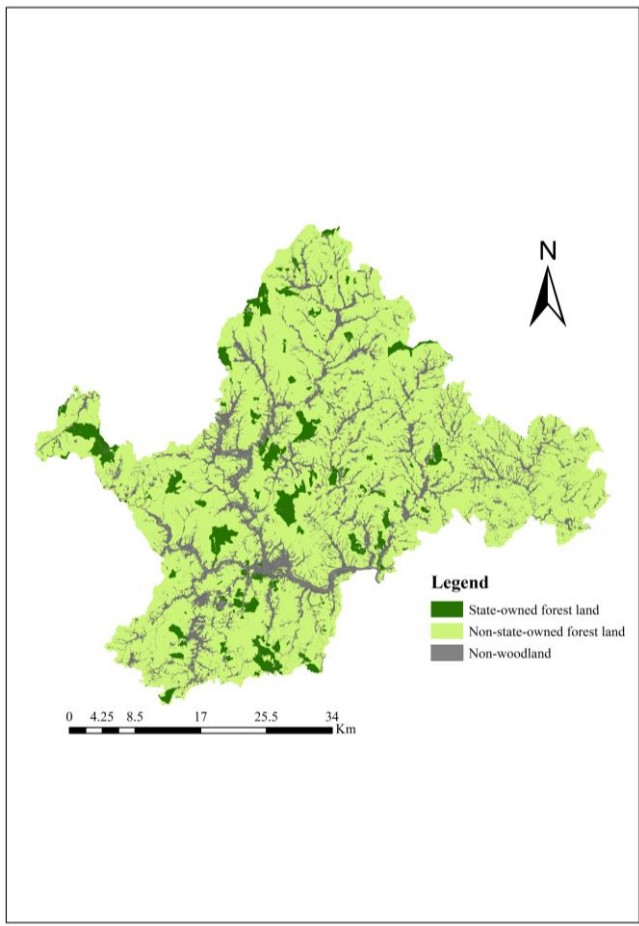

**Figure 2.** Distribution map of Shunchang County: differences in ownership of forest land.

### 3.2.2. Results of the Assessment of the Baseline Value of Forest Land

The benchmark price of forest land is the price of forest land assessed at a specific point in time; it is time-sensitive and needs to be updated regularly. Most of the updating periods for the benchmark prices of various types of land are 3 years; therefore, the benchmark price of forest land is consistent with that of other land and is updated once every 3 years. In this study, the benchmark date of the assessment was set as 1 January 2021 based on the time of the acquisition of the economic parameters. In this study, level 1 is taken as the standard grade of forest land, and the average value is taken as the base price of standard forest land after assessing and estimating the economic value of its resource assets. The land price valuations of the two forest land types are quite different, with a difference of USD 131.06 per unit area. By calculating the value-added premium value of the FEB in accurately improving forest quality, it is determined to be USD 30.19 million in Table 6.

**Table 6.** Shunchang County: woodland forest base price difference results.

| Forest Land Class | Value of State Owned per sq.km | Value of Non-State Owned per sq.km | Difference |
|---|---|---|---|
| 1 | 378.30 | 247.23 | 131.07 |
| 2 | 360.28 | 235.46 | 124.82 |
| 3 | 343.13 | 224.25 | 118.88 |
| 4 | 326.79 | 213.57 | 113.22 |

### 3.3. Results of the Realization Degree of the Ecological Product Value of the FEB

After calculation, the degree of the ecological product value realization of Shunchang County's FEB is (8221.85 + 2081.32)/12,049.62 = 85.51%. The calculation results show that the degree of ecological product value realization under the Shunchang County FEB mode is high, although less than 1, but the conversion rate is faster, indicating that the Shunchang County FEB is able to basically complete the conversion of ecological value realization into the degree of ecological product value realization and that the scale operation of the FEB through the storage of individual forest farmers' forest resources can effectively improve the quality of forests. It also actively develops forestry characteristic industries and explores a new mode of forestry carbon trading, which fully drives the transformation and development of the forest land industry.

## 4. Discussion

According to the value realization degree of the ecological products of Shunchang County's FEB, it is not difficult to find that its transformation degree is high but still not more than 1. The reason for this is that the FEB can better play the ecological benefits of forest resources, but in the process of an ecological to economic transformation, there are still some obstacles. According to the current situation of China's forestry economy, it can be divided into forestry industry development, forestry ecological product monitoring, forestry product confirmation, and so on. In order to better carry out sustainable development, we discuss the above aspects and provide corresponding suggestions to improve the realization of the value of ecological products.

### 4.1. Promoting the Integrated Development of Forest Ecological Industries

Shunchang County's FEB should be based on the characteristics of its forest resource endowment on the basis of continuously improving the quality of forest resources, fully excavating and broadening the path to realizing forest resource industrialization, promoting the integrated development mode of forest ecological industry, diversifying the revitalization of forest resources, realizing the value of forest ecological products, and continuing to promote the economic development of Shunchang County to reduce the development gap with other counties of Nanping City [32]. First, we explored the value-added model of forest ecological products. We learned from the commercial financial model, steadily expanded

the type and scale of green financial instruments, introduced financial instruments such as insurance, futures, and bonds in a timely manner, explored the establishment of characteristic agricultural insurance, forest insurance, and green bonds, and enhanced the ability to raise funds and serve entities [33]. Based on the improvement of the ecosystem's quality, we should focus on the entry of environmentally sensitive enterprises into Shunchang County to promote local economic transformation [34]. The FEB creates a good ecological environment by integrating forest resources and improving forest quality. In addition to the simple supply of material resources, it also greatly increases the income of residents and the regional economy through the spillover of economic and ecological value. A higher environmental level can attract enterprises with higher environmental sensitivity to settle in. Secondly, we should increase government intervention and improve resource allocation. The government should improve the policy bias towards rural areas, effectively regulate the allocation of resources, optimize the allocation of urban and rural resources, vigorously develop the forest industry, and transform the advantages of forest resources in Shunchang County into the efficiency of common prosperity. We should improve the forest ecological compensation mechanism, the forest ecological industry market promotion mechanism and the ecological industry development benefit sharing mechanism, broaden the forest ecological product value realization path, actively absorb the rural labor force employment, and improve the rural non-agricultural employment income and forest resources.

### 4.2. Innovating Industrial Transformation and Promoting Industrial Integration

First, we should follow the trend of the times and utilize opportunities for the development of the digital economy [35]. We should vigorously cultivate forest product industry clusters with local characteristics and brand effects, attract external investments while shortening the industrial chain, promote the return of market capital and technology, promote non-farm employment for the rural population, and consolidate and enhance the efficiency of realizing the value of forest ecological products. Secondly, it is necessary to strengthen the deep integration of forestry industries one, two, and three, accelerate the creation of the whole industry chain of forest ecological products, and increase the development of forestry industries two and three. It is important to promote the development of forest recreation, forest tourism, and other eco-enrichment industries, strengthen the application of technologies related to the industrialization of forest resources and the enhancement of operation and management capabilities, promote the conversion of forest ecological benefits for the public good from ecological benefits into economic benefits, and improve the efficiency of realizing the value of forest ecological products [36].

### 4.3. Clarifying and Improving the Registration of Ecological Products

It is necessary to identify the rights of forest ecological assets in Shunchang County, clarify the corresponding tradable ecological products and their functional and value quantities, explore the pricing model for regulating service products, and develop a market trading mechanism for regulating service products. First, we must establish a list of ecological industrialized products. Tradable property rights are the basic guarantee and fundamental guidelines for eco-industrialization, and the biggest obstacle in current practice is that the trading object is unclear [37], so it is urgent to establish an eco-industrialization product list according to the needs of eco-industrialization and include eco-products in line with regulations, industry needs, and accurate and less controversial trading on the list so as to clarify the various types of tradable eco-products of natural resource assets. On this basis, the first thing that must be clarified is the main body of natural resource ownership agents, management subjects, and users and the boundaries of their respective rights and responsibilities, and we must innovate forms of the realization of universal ownership and collective ownership of natural resource assets. Secondly, it is necessary to enrich the ownership rights of various types of natural resources and flexibly adjust the allocation of natural resource assets [38]. Enterprises, institutions, market players, and other entities should be encouraged to integrate state-owned and collective resource assets for industrial-

scale operation through franchising, authorized operation, leasing, and other paid uses, accelerate the promotion of natural resources rights registration, clarify the connotation of various types of natural resources, boundaries, and unified control rules and reduce the risk of enterprise policy. Thirdly, they have formulated differentiated environmental standards for industrial access, guided the orderly development of ecological resources, and promoted the synchronization of economic reproduction and ecological reproduction. It is necessary to combine the use control of natural ecological space with the National Ecological Civilization Pilot Area, the pilot project of unified rights registration, and the system of compensation for ecological and environmental damages, etc., and improve the use control of natural ecological space in terms of control positioning, control mode, control basis, and control method, etc., so as to form a replicable and popularized experience of the use control of natural ecological space and provide a practical basis for the establishment of a use control system for natural ecological space in the country [39]. It is necessary to provide a practical basis for the establishment of a natural ecological space use control system throughout the country.

### 4.4. Strengthening the Monitoring of Ecological Products

The monitoring of ecological products is a fundamental guarantee to promote the realization of the value of ecological products. Judging the strength of the supply capacity of ecological products and the degree of value realization can be achieved through the dynamic monitoring of ecological products. The dynamic monitoring of forest ecological products should be based on the existing forest natural resource survey system and the data of ecological positioning observations and research stations. The monitoring purpose is to determine the quantity and quality of forest ecological products and to formulate a classification system and catalogue list of ecological products. In addition, because the FEB can scientifically cultivate forests, the dynamic monitoring of forest ecological products is also continuous monitoring of forest health to continuously monitor the quality of forest products. Since the establishment of the FEB, the forest volume of Shunchang County has increased by more than 1.2 cubic meters per mu per year. The service functions of the forest ecosystem, such as water conservation and air purification, have been continuously improved. High forest quality is the material basis for the supply of ecological products. Through scientific management and large-scale and professional management, the quality of forest resources, the asset value, and the carrying capacity of a forest ecosystem can be continuously improved. Strengthening the monitoring of ecological products can reduce the problems of natural landscape destruction and biodiversity reduction caused by human intervention in forest management by the FEB. Effective monitoring can pay attention to the changes in forest ecosystems and guide timely and correct responses [40].

### 4.5. Limitations

This paper still has limitations. (1) The FEB's collection and storage area is 0.51 million hectares, but in a dynamic process, with the passage of time, the collection and storage area will continue to increase. In research that has not yet been carried out, the dynamic evaluation and analysis of time and space will be added according to the actual situation so that the research results have a control group and can provide support for decision-making managers. (2) Since our analysis of the value realization degree of the FEB's ecological products led to a result of less than 1, we can use the annual values over the last rotation as a prior distribution and randomly sample 1000 times to generate a distribution of value realization in the unexplored studies to explore when FEB operation can increase the degree of realization to 100%. (3) In our discussion, we put forward a future work improvement direction for the FEB but ignored the impact of economic situation changes, which is also an improvement direction for follow-up research. However, in this paper, changes in the economic situation will not affect the main conclusions of this paper.

## 5. Conclusions

This paper constructs an evaluation framework for the value realization degree of the ecological products of the FEB in Shunchang County by analyzing the forest land premium effect of the FEB in Shunchang County from the perspective of the ecological quality improvement of the FEB in Shunchang County. According to the analysis of the obtained research data, the production value of the ecosystem within the business scope of Shunchang County's FEB was increased to USD 25.92 million due to the accurate improvement of forest land quality. The benchmark price of state-owned forest land in Shunchang County is USD 378.30, and the benchmark price of non-state-owned forest land is USD 247.23. The value-added premium value of forest land is USD 30.19 million. Finally, this paper takes the ratio of the sum of the material supply value of the state-owned forest ecosystem and the premium benefit of the forest land to the total production value of the state-owned forest ecosystem in Shunchang County as a quantitative index of the realization degree of the ecological product value of the FEB in Shunchang County. The value realization degree of the ecological products of the FEB in Shunchang County is 85.51%. The results show that Shunchang County's FEB has a great effect on the realization of the value of ecological products. However, due to the huge ecological benefits generated, and there are still constraints on the realization of the value of forest ecological products in the development of the forestry economy, and the degree of conversion is not 100% when it is transformed into economic benefits.

At the same time, this study also provides some policy implications. First, the evaluation of the value realization degree of ecological products can accurately identify the supply capacity of ecological products, measure the gap between the value realization results of ecological products and the established goals, and help the government find flaws in ecological protection in the management process. Second, the evaluation system we constructed can reasonably calculate the economic value and ecological value of forest quality in the region and can be effectively used to help the government calculate the value of ecological assets. Third, it is helpful for the government to explore a new realization path based on the existing ecological product value realization and provide data support for future sustainable development.

**Author Contributions:** Conceptualization, D.X., L.W. and W.Z.; methodology, D.X., L.W. and W.Z.; validation, D.X., L.W. and W.Z.; formal analysis, D.X. and L.W.; investigation, D.X. and L.W.; resources, W.Z.; writing—original draft preparation, D.X., Y.W. and L.W.; writing—review and editing, D.X., Y.W. and W.Z.; visualization, D.X. and L.W.; supervision, W.Z.; project administration, W.Z.; funding acquisition, W.Z. All authors have read and agreed to the published version of the manuscript.

**Funding:** This research was funded by the Nanping ecological product value realization theory method and system construction research project, grant number SK220432.

**Data Availability Statement:** The data presented in this study are available on request from the corresponding author. The data are not publicly available due to obtain from the author's field research.

**Conflicts of Interest:** The authors declare no conflict of interest.

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
