# Peer review of "Evaluation of the Degree of the Value Realization of Ecological Products of the Forest Ecological Bank in Shunchang County"

_forests, doi:10.3390/f14112269_

Round 1
Reviewer 1 Report
Comments and Suggestions for Authors
Dear Authors,
I have reviewed the paper titled: “Research on the evaluation of the value realization degree of ecological products of forest ecological bank in Shunchang County". In my opinion, the aims of the paper are germane with “Forests” journal topic, however in the present form, the paper has some important flaws. The contribution of this paper to the scientific knowledge is good. I understand the difficult work done, but as a reviewer it is my duty to highlight the gaps in order to improve the research approach and its presentation to the international scientific community. Please, in order to have a good possibility for a possible future publication I attached the pdf file with some important comments. Further I suggest you to make a better linkage among aims and conclusions.

Some minor errors are still present, I suggest a further check.
Reviewer 2 Report
Comments and Suggestions for Authors
The topic of the research work and manuscript is really interesting and provides new information. However there are some issues to be addressed towards its quality improvement before publication.
Since the journal is international and the readers from all around the world, it would be a good idea to provide in the text the values both in yuan and euro or dollars. Do not leave space between the value and the percentage symbol. In the results discussion area, I am not sure about the use of word innovate.It does not seem to be suitable in the way it is used (the meaning is not clear). In line 47, in "The creation of the forest ecological bank has realized the unity..", the verb does not seem to be right or the meaning should be clarified in another way. The introduction is much generic and it does not present the so far literature review results of the authors, the state-of-the-art analysis (where such forest management practices have been applied and which were the results etc.). I would recommend the authors to highlight in the text as well the need for continuous monitoring of forest health and therefore, the quality of forest products, using a relevant study (DOI: 10.1109/ICASSP.2019.8683128 ) to support your statements. In the abstract, as well as in the end of introduction and in conclusions chapter, the significance and practical meaning of this work is not highlighted, not even referred. How the results are going to be utilized in the near future? It is clear that all the bibliography, all the references used are all from local sources of the same nationality of the authors'. That would not be a significant problem, if you had taken into consideration as well practices from all around the globe, though such a thing (international literature review) is at least questionable. In discussion area you should provide a brief comment on the anticipated results of such forest bank practice in the next years concerning the protection and sustanability of virgin high-value forest ecosystems. Is it possible to loose their character and unique appearance, biodiversity and balance?
Comments on the Quality of English LanguageThe use of English language is in general satisfying, through there are some points that need to be improved in order to clarify the meaning. Please, check as well the comments to the authors. I remain at your disposal for any clarification.
Reviewer 3 Report
Comments and Suggestions for Authors
Forests-2688058 “Research on the evaluation of the value realization degree of ecological products of forest ecological bank in Shunchang County” Ding Xu, Lianbei Wu and Weimin Zhang
For an article that is predominantly an economics venture, I was a little disappointed to get to the end of Section 3 and see the answer is 85.5%. That was it. It all boiled down to a single value. Section 4 was an interesting read, but I would have liked a little more analysis. Perhaps the use of a distribution of input costs and interest rates might have given some idea of the possible range of value realization rather than a single value. How might things change is we were in the low interest rate times of the pandemic, or long-term high interest rates at current levels, or the variability in the price of forest products. Maybe use historical input prices, interest rates and investment costs to generate a different value over a single rotation. Why not use the annual values over the last rotation as a prior distribution and randomly sample 1000 times to generate a distribution of value realization? There will then be a probability that under some of the historical conditions, if considered as average over the entire rotation, that the value realization is more than 100%.
Comments below are editorial in nature.
· The title is awkwardly worded and could be changed to be consistent with the Abstract to “Evaluation of the degree of value realization of ecological products of forest ecological bank in Shunchang County”. Perhaps “forest ecological bank” needs the acronym FEB in the article, as the phrase sometimes has capital letters, sometimes none, sometimes is in quotation marks, and is otherwise inconsistent throughout the text.
· I doubt that “green hills are golden hills” is a scientific theory as we commonly understand it (Introduction, line 37). The explanation that follows does a good job of supplying context to this distilled slogan, however it could be better expressed as: “China is building on the concept that “Green hills are golden hills”.”
· There are too many different units of measurement used in the article. There are SI square kilometres, Imperial acres, and standard Chinese mu (approx. 666.7 square metres). For an international audience, please convert all areas to SI units of metres or kilometres. Currency also switches between “yuan” and “RMB”; apparently CNY is the ISO 427 standard abbreviation and one of these should be used consistently throughout the text.
· Figure 1 needs to show Shunchang County within a map of China so the reader can easily locate it, rather than a zoomed in picture of its outline. As it is only small, a portion of southern China with Shanghai in the top-right, Hong Kong in the bottom-left, and the Fujian provincial boundary should be enough for context.
· The first paragraph of §2.2.1 has many words that are mixed in the sentences and does not make complete sense. Try: “The production value accounting of Shunchang forest ecosystem is based on China's forest ecosystem service function assessment specification [12-13]. The difference between the production value per unit area of state-owned and non-state-owned forest land, combined with the ecological bank storage value, is the ecological value enhancement impact brought by the operation of Shunchang County Forest Ecological Bank to Shunchang County. The production value of forest ecosystems accounts for the sum of the values of all ecological goods and services provided by ecosystems to human society in a given region, and is expressed as:”
· Table 1 would benefit from some horizontal lines between secondary indicators so the reader can see which characteristics and indices are related to each indicator.
· Please number all equations, starting on page 3 line 108. The font size on second equation (page 6 line 197) is much larger than the three that follow and there does not appear to be a good reason given the similarity between equations on lines 197 and 199.
· In Roman Numerals it is common to show the number 4 as “(iv)” (page 7 line 212) and the number 5 as “(v)” (page 7 line 219).
· The ellipses in the equation of Bu (benchmark value of forest land) on page 7 line 229 indicate there are other Dj terms not listed. Why isn’t this also presented as a sum like the Cj terms? If some Dj terms are zero because of harvesting schedules then this is fine, and would also allow both sets of annual terms to be combined into a single sum of (Dj – Cj) with the common factor (1+i)(u-j+1).
· Need to remove a hanging phrase “The present value of G” on page 8 line 257.
· Remove all the decimal places from values in Tables 2 and 3. Do the Physical or Value Quantities in Table 2 have units? If so, they should be indicated in the row headers. Are there really 20 orders-of-magnitude difference in value quantities? If so, then you need to find some better way to express these quantities, but I suspect the entry for “negative oxygen ion release” is an error. You do not need to have “Shunchang” as the first column in Table 3 as it is redundant.
· Tables 6 and 7 can be combined as the Number of Classes and Area columns are already shown earlier. The new Table 6 would only have columns for Land Class, Value of State Owned per sq.km, and Value of non-State Owned per sq.km, with a fourth column containing the difference if that seems important.
· Something odd is happening in the References where five occurrences of “Xi” are replaced with “**”, in both people’s names and places, and needs to be corrected.
Comments on the Quality of English Language
Some sentence structure is awkward, with the major offenses outlined in comments. The authors also tend to repeat the first few words of sentences as the last few words, which is unnecessary.
Reviewer 4 Report
Comments and Suggestions for Authors
The issue addressed in the paper discusses the value realization degree of ecological products of forest ecological bank in Shunchang County.
First of all, I find that an important topic, compatible with the journal's scope, was considered. please note, that some such studies are partially analysed in literature. It would be worth presenting the state of the art in a broader way. I suggest a more dilligent, comparative description of other scientific research from the literature (please complete section 1).
I also recommend several corrections to improve the quality of this paper:
- to improve the readability and description of figures (since they are the basis for analysis verification), supplement the history of their description, a clear and not laconic reference in the paper (in section 2: figure 1, in section 3: figure 2);
- - to clarify whether it is also possible to use another approach, other indicators (please justify the use of just such a measure in the study: “quantitative index of the realization degree of the ecological product value of the forest ecological bank in Shunchang County”).
- - to be detailed (in direct connection with the main questions of the study): discussion of results (in section 4).
Please remember that the formulated objectives – should find a clear answer in the study conclusion. Is this really the way it works? Does the conclusion answer all the questions posed at the beginning of the paper (expressed in objectives and hypotheses)? It is also a great idea to present these most important [already sorted, synthetic] results in a clear way.
I also strongly suggest that recommendations for specific, practical, not only general (and not entirely clear) applications of this research shall be provided (section 5).
Comments on the Quality of English LanguageThe language of this paper is relatively correct, however some descriptions would benefit from being more concise.
Round 2
Reviewer 1 Report
Comments and Suggestions for Authors
Dear Authors,
I have reviewed the paper titled: “Evaluation of the degree of value realization of ecological products of forest ecological bank in Shunchang County". In my opinion, as stated in the previous step the aims of the paper are germane with “Forests” journal topic, however also in the present form, the paper remains with some important flaws. The contribution of this paper to the scientific knowledge could be good but the major part of my previous comments have not been addressed or correctly addressed. The paper still lacks of a clear presentation of the research hypotheses (done in part) or questions and of a proper study limitation sub-section in the discussion (not clear). Also, minor comments like the one regarding the first sentence of the manuscript were not addressed. I understand the difficult work done, but as a reviewer it is my duty to highlight the gaps in order to improve the research approach and its presentation to the international scientific community.
Comments on the Quality of English LanguageOnly minor mistakes, please make a further english grammar check.
Reviewer 2 Report
Comments and Suggestions for Authors
As I have checked the authors have implemented the proposed changes in the revised verion of manuscript towards the improvement of their work. Almost all the changes have been implemented and in my opinion, the manuscript is well-prepared and organized enough to be accepted for publication in this journal. I remain at your disposal for any clarification.
Author Response
Thank you very much for reviewing our article. Your every suggestion is very valuable to us, thank you.